# Willingness to get vaccinated initially and yearly against COVID-19 and its association with vaccine hesitancy, vaccine knowledge and psychological well-being: a cross-sectional study in UK adults

Dimitra Kale ![ORCID],[1] Emily Shoesmith,[2] Aleksandra Herbec ![ORCID],[1,3] Lion Shahab[1]

¹Department of Behavioural Science and Health, Institute of Epidemiology & Health Care, London, UK
²University of York, York, UK
³Institute–European Observatory of Health Inequalities, Calisia University, Kalisz, Poland

Correspondence to
Dr Dimitra Kale;
dimitra.kale.09@ucl.ac.uk

## ABSTRACT

**Objectives** This study explores the association between vaccine hesitancy, vaccine knowledge and psychological well-being with (1) receipt of/willingness to receive an initial vaccine against COVID-19, and (2) willingness to get vaccinated yearly against COVID-19. The importance of different vaccine attributes (eg, vaccine technology, effectiveness, side effects) to choose a specific COVID-19 vaccine was also assessed.

**Design** Cross-sectional survey administered during May to June 2021 on vaccine hesitancy, vaccine knowledge, psychological well-being, willingness to receive COVID-19 vaccines, sociodemographics and COVID-19-related factors.

**Setting** UK.

**Participants** A self-selected sample of 1408 adults.

**Outcome measures** Receipt of/willingness to receive COVID-19 vaccine for the first time and yearly.

**Results** Receipt of/willingness to receive a vaccine against COVID-19 initially and yearly were high (97.0% and 86.6%, respectively). Vaccine hesitancy was negatively associated with receipt of/willingness to receive vaccine initially/yearly (adjusted OR (aOR)=0.09, 95% CI 0.04 to 0.26, p<0.001/ aOR=0.05, 95% CI 0.03 to 0.09, p<0.001). Vaccine knowledge and psychological well-being were positively associated with willingness to receive a yearly vaccine (aOR=1.81, 95% CI 1.43 to 2.29, p<0.001 and aOR=1.25, 95% CI 1.02 to 1.51, p=0.014, respectively), and general vaccine knowledge also with receipt of/willingness to receive vaccine initially (aOR=1.69, 95% CI 1.18 to 2.42, p=0.004). Vaccine effectiveness was the most important attribute for participants to choose a specific COVID-19 vaccine.

**Conclusions** Improving vaccine knowledge and emphasising vaccine efficacy may minimise vaccine hesitancy and increase COVID-19 vaccine uptake.

## INTRODUCTION

COVID-19, caused by SARS-CoV-2, is one of the deadliest communicable diseases of the 21st century,[1] which led to a pandemic that burdened all aspects of life worldwide.[2]

## STRENGTHS AND LIMITATIONS OF THIS STUDY

⇒ The study benefited from a large sample size.
⇒ The use of validated measures of vaccine hesitancy and vaccine knowledge.
⇒ Examining willingness to get vaccinated against COVID-19 during the roll-out of the first COVID-19 vaccine programme, which improves the validity of findings and minimises recall bias.
⇒ Use of a convenience sample, which may not be representative of the general population.
⇒ Self-reported data and thus susceptible to social desirability bias.

To contain the pandemic, vaccines against COVID-19 were quickly produced and approved for use in the general population in late 2020 and early 2021 across different countries. The development of the COVID-19 vaccines unfolded in real time, lending a sense that they were developed more rapidly than other vaccines; some used new technology to trigger an immune response (ie, mRNA technology), raising concerns regarding their safety and effectiveness.[3] Subsequently, these concerns have increased COVID-19 vaccine hesitation, a delay in acceptance or refusal of vaccination despite the availability of vaccination services,[4] around the world.[5] Understanding correlates of COVID-19 vaccine acceptance and vaccine attributes that are important for people when considering getting a COVID-19 vaccine (ie, vaccine technology) is therefore critical to designing a successful immunisation programme that will continue to contain the COVID-19 pandemic and prevent potentially devastating outbreaks in the future. Also, it will help with the development of

evidence-based interventions to address vaccine hesitancy in general, which further increases the importance of research in this area.

A growing literature studies the determinants of vaccine hesitancy related to COVID-19 and willingness to get vaccinated against COVID-19. Factors associated with willingness to get vaccinated against COVID-19 include multiple sociodemographics, behavioural and psychological correlates, as well as pandemic-related factors.[6–16] Specifically, it has been shown that being older, male, of white ethnicity, with high income and post-16 qualification have been consistently related to higher COVID-19 vaccine acceptance and lower COVID-19 vaccine hesitancy.[6–10] Additionally, a positive correlation has been reported between willingness to get vaccinated against COVID-19 and negative affect, with those reporting more worry and anxiety related to COVID-19 also reporting more willingness to get vaccinated against COVID-19.[11 12] However, some data suggest that poor psychological well-being and significant levels of depression are associated with vaccine hesitancy.[13] Pandemic experiences, such as history of COVID-19 infection, and higher perceived likelihood of getting COVID-19,[9–11 15 16] as well as vaccine-related attributes such as efficacy and side effects[17] have also been associated with higher COVID-19 vaccine acceptance. However, most of these studies were assessing hypothetical vaccine acceptance and hesitancy.

Most COVID-19 vaccination regimes require at least two doses to reach the desired immunisation level, and additionally many countries encouraged and facilitated booster doses among its population.[18] Examining rates of COVID-19 vaccination, it seems that they were higher for the first dose of COVID-19 vaccine in comparison with follow-up/booster doses. For example, in the UK, the prevalence of receipt of the first COVID-19 vaccination dose was 93.6%, while 88.3% and 70.2% for the second and the third dose, respectively.[19] These rates confirm earlier research, suggesting that a considerable proportion of the general population who have received two doses of a COVID-19 vaccine were either unwilling or unsure about accepting a booster vaccine.[12 13] Factors associated with this unwillingness and uncertainty include lower levels of stress about catching or becoming seriously ill from COVID-19, lower levels of educational qualification, lower socioeconomic position and younger age.[12]

Emerging data also consistently show that vaccine effectiveness against COVID-19 infection declines over time.[20] Thus, administering additional COVID-19 vaccine doses to appropriate individuals (ie, elderly, those with vulnerabilities, healthcare professionals) regularly may be necessary to protect susceptible individuals against hospitalisation and death. Indeed, similar to influenza vaccines, the COVID-19 vaccine autumn boosters have been offered in the UK to vulnerable groups and healthcare professionals.[19] Officials in the USA and the UK have suggested that we will need to include annual COVID-19 vaccinations, with a schedule similar to influenza shots to prevent potentially devastating outbreaks in the future.[19 21]

Crucially, the effectiveness of such programmes depends on vaccine uptake.

While several studies have sought to identify factors associated with willingness to get vaccinated against COVID-19, research examining people's willingness to get vaccinated against COVID-19 on a yearly basis is scarce. For example, Pal *et al*[22] examined attitudes towards a hypothetical annual booster vaccine against COVID-19 in US healthcare workers, and they reported that overall acceptance was 83.6%.[22] However, this acceptance was widely divergent among the vaccine-hesitant and non-hesitant groups (13.8% vs 89.9%).[22] Moreover, most studies were conducted prior to the approval of a COVID-19 vaccine.[9–16] Thus, it is not known whether these factors have the same importance when the outcome is real rather than hypothetical. Data for the current study were collected during the first COVID-19 vaccine roll-out (May to June 2021), and therefore our participants were asked about their willingness or acceptance of the initial COVID-19 vaccine, and willingness of a yearly COVID-19 vaccination while they had been already offered the first dose of COVID-19 vaccine or were likely to be offered it soon.

The aim of this study is to explore whether there is an association between vaccine hesitancy, general vaccine knowledge (eg, the process related to vaccination, the impact of vaccination) and psychological well-being during the COVID-19 pandemic with (1) receipt of/willingness to receive vaccine against COVID-19 for the first time, and (2) willingness to get vaccinated against COVID-19 on a yearly basis in a sample of UK adults during the period of the first COVID-19 vaccine roll-out. Greater understanding of the associations between vaccine hesitancy, vaccine knowledge and psychological well-being with willingness to get vaccinated against COVID-19 could help us design mitigating strategies or interventions going forward (ie, educational or psychological support). We also assess to what extent different vaccine attributes (ie, vaccine technology, effectiveness, side effects) are important for people to choose a specific COVID-19 vaccine, if they had the choice, which may also indicate attributes that are important for getting a vaccine against COVID-19.

### Research questions

Research question 1 (RQ1): Is there an association between (1) vaccine hesitancy, (2) vaccine knowledge and (3) psychological well-being during the COVID-19 pandemic with receipt of/willingness to receive an initial vaccine against COVID-19 in a sample of UK adults during the period of the first COVID-19 vaccine roll-out, adjusting for relevant covariables?

RQ2: Is there an association between (1) vaccine hesitancy, (2) vaccine knowledge and (3) psychological well-being during the COVID-19 pandemic with willingness to get vaccinated against COVID-19 on a yearly basis in a sample of UK adults during the period of the

first COVID-19 vaccine roll-out, adjusting for relevant covariables?

RQ3: Which attributes (ie, vaccine technology, effectiveness, side effects) are most important to people when considering getting a COVID-19 vaccine, if they had the choice?

## METHODS

### Study design

This is a cross-sectional study using data of a longitudinal online survey of adults residing in the UK; the HEalth BEhaviours during the COVID-19 pandemic (HEBECO) study (https://osf.io/sbgru/). Baseline data collection occurred between April and June 2020, and follow-up surveys were administered at 1 month, and 3, 6 and 12 months from the baseline participation date. The current study uses data from the baseline and 12-month follow-up surveys.

### Patient and public involvement

At the end of every survey, we asked participants to provide comments related to the content of the surveys, which was considered when the following survey was created. There was no patient and public involvement in the design of the baseline survey, but participants' comments on baseline, 1 month, and 3 and 6-month surveys were considered when designing the 12-month follow-up survey. Participants and the public can access study results at the study website, which is freely available.

### Study sample

A self-selected sample of UK-based adults (18+ years) who completed both the baseline survey of the HEBECO study between 23 April 2020 (initiation of recruitment) and 14/June/2020 (inclusive; marking the end of the first national UK lockdown), and a 12-month follow-up survey collected between May and June 2021. At the time of the 12-month follow-up survey, all clinically extremely vulnerable individuals and those aged 40+ had been offered at least one dose of COVID-19 vaccine in the UK.

Initial recruitment into the baseline survey was online and involved sharing study invitations via multiple channels, including unpaid and paid advertisements on social media (eg, Facebook), an email campaign across the network of University College London, other universities in the UK, Public Health England, Cancer Research UK, charities and local authorities across the UK. The full recruitment strategy is available online (https://osf.io/sbgru/).

Participants gave their consent prior to data collection. Data were captured and managed by the REDCap electronic data system.[23 24] Participants were followed up at 12 months after their baseline survey via email (except for participants who explicitly opted out), with up to three reminders to complete the survey sent at 12-month follow-up.

## Measures

The questionnaires were developed by a multidisciplinary team of academics with input from Cancer Research UK and Public Health England (https://osf.io/sbgru/). The questionnaires included validated items and new items based on expert consensus relating to emerging COVID-19-related aspects, as detailed below. All measures were self-reported.

### Outcomes (assessed at 12-month follow-up)

Receipt of/willingness to receive vaccine against COVID-19 for the first time. As COVID-19 vaccines became available in mid-December 2020 in the UK, at 12-month follow-up (May to June 2021), individuals reported whether they had been offered a vaccine for COVID-19 (yes/no). Those who replied yes were asked if they had received a COVID-19 vaccine with the following answer options: (1) yes, both doses; (2) yes, one dose; (3) no, but I am scheduled to receive the vaccine; (4) no, I decided not to get vaccinated; (5) prefer not to say. Participants were classified as willing to receive/received the vaccination if replied (1), (2) or (3), and not willing if they selected (4) and (5). This measure was structured by researchers and was based on previous assessments of COVID-19 vaccine.[25]

Willingness to get vaccinated against COVID-19 on a yearly basis was assessed with a question 'would you be willing to get vaccinated against the novel coronavirus (SARS-CoV-2) on a yearly basis?' with the following answer options: (1) yes, (2) no, (3) not sure. Willingness to get vaccinated against COVID-19 on a yearly basis was conceptualised as a binary variable, yes versus no/not sure. We also ran sensitivity analysis where willingness was treated as a three-category variable, yes/no/not sure.

Importance of COVID-19 vaccine attributes was assessed by asking participants to rank in order of importance for them, if they had the choice to select a specific COVID-19 vaccine, the following six attributes: (1) vaccine technology (established vs novel), (2) level of protection (how effective it is), (3) side effect profile (how easy it is tolerated), (4) delivery method (injected or not, one dose or more), (5) ease of access (walk-in centres vs general practitioner appointment), (6) reputation (manufacturer, media coverage). Participants could rank the attributes from 1=most important to 6=least important, and they could not assign the same ranking to more than one attribute.

### Explanatory variables (assessed at 12-month follow-up)

Vaccine hesitancy was assessed using a 9-item scale,[26] revised from the 10-item vaccine hesitancy survey tool developed by the SAGE Working Group on Vaccine Hesitancy.[27] The original scale assesses parental attitudes about childhood vaccines, while this adapted version focused on the respondents themselves and is validated in the UK population.[27] Participants were asked to answer nine questions related to their confidence in vaccines in general, rather than vaccines for COVID-19 specifically,

on a 5-point Likert scale (1=strongly agree to 5=strongly disagree). A continuous score was calculated, with higher scores reflecting a higher level of vaccine hesitancy. Cronbach's alpha values in the present study were 0.70.

Knowledge about vaccines was assessed using a 9-item scale that includes questions about the immunisation process related to vaccination, the impact of vaccination and the consequences of vaccination in general, rather than vaccines for COVID-19 specifically. This scale has shown good psychometric properties.[28] Answer options were dichotomised into 1=correct and 0=incorrect/don't know. A continuous score was calculated, with higher scores presenting a higher knowledge about vaccines. Cronbach's alpha values in the present study were 0.61.

Psychological well-being during the COVID-19 pandemic was assessed with the question 'How would you rate psychological well-being' measured on a scale from '1=poor' to '5=excellent'. It was treated as a continuous variable with higher scores presenting better psychological well-being.[29]

## Covariables

Sociodemographic characteristics were assessed at baseline and included age (continuous in years), gender (female vs all other), education (post-16 qualification vs not), ethnicity (any white ethnicity vs all other/prefer not to say) and health conditions (any health condition vs none/prefer not to say).

COVID-19-related characteristics were assessed at 12-month follow-up and include COVID-19 risk to one's health (major risk/significant vs moderate/minor/no risk at all/don't know), isolation status (total vs some/general/no isolation), diagnosed/suspected COVID-19 (yes vs no/prefer not to say) and worry about future COVID-19 waves (continuous; 0=not at all, 50=neutral, 100=extremely, higher scores representing more worry).

## Confounder variable

Timing (day) of the 12-month follow-up survey completion (daily basis; continuous) was included to control for changes in willingness to get vaccinated in light of the changes in social distancing measures in the UK during the period of data collection.

## Statistical analysis

Descriptive statistics were calculated to characterise the sample. Data were analysed with SPSS V.27 on complete cases only. The protocol and analysis plan were pre-registered on Open Science Framework (https://osf.io/s46qm). To account for the non-random nature of the sampling, data were weighted to the proportions of sex, age, ethnicity, education and country of living obtained from the Office for National Statistics (2018) for the descriptive analysis. Data were unweighted for the inferential analysis.

Logistic regressions for unweighted data were conducted to examine the association between (1) vaccine hesitancy, (2) vaccine knowledge and (3) psychological

well-being during the COVID-19 pandemic and (RQ1) receipt of/willingness to receive an initial vaccine against COVID-19 and (RQ2) willingness to get vaccinated against COVID-19 on a yearly basis. We run both unadjusted and adjusted analyses controlling for relevant covariables and timing of 12-month follow-up survey. To identify the relevant covariables for each model, we drew directed acyclic graphs (DAGs) to make the assumptions about the relationships between variables explicit.[30] We used an online tool, DAGitty v3.0, to create the DAGs (http://www.dagitty.net/). Variables and their measurements are depicted by nodes and connected by unidirectional arcs depicting the hypothesised relationship. For RQ1 models, our DAG (online supplemental figure 1) implied that the following confounding variables required conditioning (as a minimal sufficient adjustment set): COVID-19 risk to one's health, age, diagnosed or suspected COVID-19, education, health conditions, isolation status and worry about future COVID-19 waves. For RQ2 models, our DAG (online supplemental figure 2) implied that the following confounding variables required conditioning (as a minimal sufficient adjustment set): COVID-19 risk to one's health, age, diagnosed or suspected COVID-19, health conditions, isolation status, receipt of/willingness to receive vaccine against COVID-19 for the first time and worry about future COVID-19 waves.

In the sensitivity analysis, multinomial regressions were conducted to examine the association between (1) vaccine hesitancy, (2) vaccine knowledge and (3) psychological well-being during the COVID-19 pandemic and willingness to get vaccinated against COVID-19 on a yearly basis, where willingness was treated as a three-category variable: (1) yes, (2) no, (3) not sure. The reference category for the outcome variable was the response option 'yes'.

For RQ3, descriptive analysis of the proportions and 95% CIs of the most important vaccine attributes for people to choose a specific COVID-19 vaccine was conducted using weighted data.

## RESULTS

Out of a total of 2992 UK adults recruited into the HEBECO baseline survey, 1565 (weighted 1408) participants were successfully followed up at 12 months and provided data on the variables of interest, and hence met the inclusion criteria for the present analysis. Comparison of those included in the present analysis and those meeting criteria for exclusion showed that included participants were more likely to be older, female, of white ethnicity and had post-16 qualifications (online supplemental table 1). The weighted characteristics of the included sample are presented in table 1. Included participants were on average in their 50s, half of them were female, most of them were of white ethnicity and two-thirds had post-16 qualifications. Just under half had a health problem, most had low-risk perceptions for COVID-19, two-thirds were worried about future COVID-19 waves, only 17 were in total isolation and one-third reported diagnosed/

**Table 1** Sample characteristics (weighted; n=1408)

| | |
|---|---|
| Age in years, M (SD) | 49.87 (16.14) |
| Female sex, % (n) | 50.4 (709) |
| White ethnicity, % (n) | 89.9 (1261) |
| Post-16 qualifications, % (n) | 68.1 (958) |
| Health problems, % (n) | 44.1 (617) |
| Perceived high risk of COVID-19, % (n) | 11.5 (161) |
| Being in isolation, % (n) | 1.2 (17) |
| Diagnosed/suspected COVID-19, % (n) | 29.7 (418) |
| Worry about future COVID-19 wave, M (SD) | 65.54 (23.36) |
| Vaccine hesitancy score, M (SD) | 1.99 (0.48) |
| Vaccine knowledge score, M (SD) | 7.29 (1.13) |
| Psychological well-being, M (SD) | 3.18 (1.07) |

Vaccine hesitancy scores range from 1 to 5, with higher score representing higher vaccine hesitancy; vaccine knowledge scores range from 0 to 9, with higher score representing higher vaccine knowledge; psychological well-being scores range from 1 to 5, with higher score representing better psychological well-being.
M, mean.

suspected COVID-19. The sample had low vaccine hesitancy, high vaccine knowledge and their psychological well-being was average.

### RQ1 results

Of the 1408 participants included in the present study, 87.2% (95% CI 85.3% to 89.0%) had been offered a vaccine for COVID-19 by the time of the 12-month survey and of these, 97.0% (95% CI 96.0% to 98.0%) had received or were willing to receive the initial vaccine against COVID-19.

Both unadjusted and adjusted analyses showed a significant negative association between vaccine hesitancy and receipt of/willingness to receive an initial vaccine against COVID-19 (OR=0.05, 95% CI 0.03 to 0.10, p<0.001; adjusted OR (aOR)=0.09, 95% CI 0.04 to 0.26, p<0.001, table 2). A significant positive association was found between vaccine knowledge and receipt of/willingness to receive a vaccine against COVID-19 for the first time (OR=2.71, 95% CI 2.17 to 3.39, p<0.001; aOR=1.69, 95% CI 1.18 to 2.42, p=0.004). No significant association was observed between psychological well-being during

**Table 2** Receipt of/willingness to get vaccinated against COVID-19 for the first time: association with vaccine hesitancy, knowledge and psychological well-being (unweighted sample)

| | Receipt of/willingness to get an initial vaccine against COVID-19* | | | | |
|---|---|---|---|---|---|
| | n=1415 | | n=1406 | | |
| | OR (95% CI) | P value | aOR (95% CI) | P value | |
| Vaccine hesitancy | 0.05 (0.03 to 0.10) | <0.001 | 0.09 (0.04 to 0.26) | <0.001 | |
| Vaccine knowledge | 2.71 (2.17 to 3.39) | <0.001 | 1.69 (1.18 to 2.42) | 0.004 | |
| Psychological well-being | 1.39 (0.97 to 1.99) | 0.072 | 0.92 (0.56 to 1.48) | 0.719 | |
| Post-16 qualifications | | | | | |
| Yes | 1 (ref) | | 1 (ref) | | |
| No | 0.83 (0.28 to 2.41) | 0.728 | 0.53 (0.19 to 1.48) | 0.227 | |
| Health problems† | | | | | |
| Yes | 1 (ref) | | 1 (ref) | | |
| No | 0.55 (0.25 to 1.21) | 0.134 | 0.80 (0.27 to 2.43) | 0.699 | |
| Perceived high risk of COVID-19 | | | | | |
| No | 1 (ref) | | 1 (ref) | | |
| Yes | 1.51 (0.35 to 6.40) | 0.580 | 1.20 (0.17 to 8.46) | 0.852 | |
| Confirmed/suspected COVID-19 | | | | | |
| No | 1 (ref) | | 1 (ref) | | |
| Yes | 0.50 (0.24 to 1.06) | 0.069 | 1.10 (0.40 to 3.03) | 0.854 | |
| Being in isolation | | | | | |
| No | 1 (ref) | | 1 (ref) | | |
| Yes | 0.20 (0.03 to 1.64) | 0.135 | 1.29 (0.04 to 38.77) | 0.885 | |
| Age (continuous) | 1.04 (1.02 to 1.07) | 0.001 | 1.09 (1.04 to 1.14) | <0.001 | |
| Worry about future COVID-19 wave | 1.05 (1.03 to 1.06) | <0.001 | 1.03 (1.01 to 1.05) | <0.001 | |
| Time of completion | 1.01 (0.96 to 1.05) | 0.843 | 1.04 (0.97 to 1.11) | 0.308 | |

*Only three participants did not disclose their preference and selected 'don't know'.
†In unadjusted analysis of health problems, n=1406, as data were missing for some of the participants.
aOR, adjusted OR; ref, reference category.

**Table 3** Willingness to get vaccinated against COVID-19 yearly: association with vaccine hesitancy, knowledge and psychological well-being (unweighted sample)

| | Willingness to get yearly vaccination against COVID-19 | | | |
| --- | --- | --- | --- | --- |
| | n=1565 | | n=1406 | |
| | OR (95% CI) | P value | aOR (95% CI) | P value |
| Vaccine hesitancy | 0.03 (0.02 to 0.05) | <0.001 | 0.05 (0.03 to 0.09) | <0.001 |
| Vaccine knowledge | 2.80 (2.37 to 3.32) | <0.001 | 1.81 (1.43 to 2.29) | <0.001 |
| Psychological well-being | 1.39 (1.19 to 1.62) | <0.001 | 1.25 (1.02 to 1.51) | 0.014 |
| Health problems* | | | | |
| Yes | 1 (ref) | | 1 (ref) | |
| No | 0.99 (0.72 to 1.36) | 0.937 | 0.98 (0.61 to 1.54) | 0.816 |
| Perceived high risk of COVID-19 | | | | |
| No | 1 (ref) | | 1 (ref) | |
| Yes | 1.57 (0.83 to 2.97) | 0.163 | 2.58 (0.99 to 6.72) | 0.053 |
| Confirmed/suspected COVID-19 | | | | |
| No | 1 (ref) | | 1 (ref) | |
| Yes | 0.53 (0.38 to 0.73) | <0.001 | 0.77 (0.49 to 1.21) | 0.256 |
| Being in isolation | | | | |
| No | 1 (ref) | | 1 (ref) | |
| Yes | 0.41 (0.11 to 1.49) | 0.173 | 0.65 (0.07 to 6.36) | 0.714 |
| Receipt of/willingness to receive an initial COVID-19 vaccine† | | | | |
| Yes | 1 (ref) | | 1 (ref) | |
| No | 0.01 (0.003 to 0.04) | <0.001 | 0.05 (0.01 to 0.26) | <0.001 |
| Age (continuous) | 1.02 (1.01 to 1.03) | <0.001 | 1.06 (1.04 to 1.08) | <0.001 |
| Worry about future COVID-19 wave | 1.03 (1.02 to 1.03) | <0.001 | 1.03 (1.01 to 1.05) | 0.008 |
| Time of completion | 0.98 (0.97 to 1.00) | 0.074 | 0.98 (0.95 to 1.01) | 0.130 |

*In unadjusted analysis of health problems, n=1554, as data were missing for some of the participants.
†In unadjusted analysis of receipt of/willingness to receive COVID-19 vaccine for the first time, n=1415, as data were missing for some of the participants.
aOR, adjusted OR; ref, reference category.

the COVID-19 pandemic and receipt of/willingness to receive an initial vaccine against COVID-19. Being older and higher levels of worry about future COVID-19 waves were significantly associated with receipt of/willingness to receive an initial vaccine against COVID-19 (table 2).

### RQ2 results
Of the 1408 participants, 86.6% (95% CI 84.7% to 88.5%) were willing to get vaccinated against COVID-19 on a yearly basis. Both unadjusted and adjusted analyses showed a significant negative association between vaccine hesitancy (OR=0.03, 95% CI 0.02 to 0.05, p<0.001; aOR=0.05, 95% CI 0.03 to 0.09, p<0.001) and willingness to get vaccinated against COVID-19 on a yearly basis. Willingness to get yearly vaccinations against COVID-19 was positively associated with vaccine knowledge (OR=2.80, 95% CI 2.37 to 3.32, p<0.001; aOR=1.81, 95% CI 1.43 to 2.29, p<0.001) and psychological well-being (OR=1.39, 95% CI 1.19 to 1.62, p<0.001; aOR=1.25, 95% CI 1.02 to 1.51, p=0.014, table 3). Being older, higher levels of worry

about future COVID-19 waves and receipt of/willingness to receive an initial COVID-19 vaccine were significantly associated with willingness to get yearly vaccination against COVID-19 (table 3).

Sensitivity analysis confirmed results, finding a significant negative association between vaccine hesitancy and willingness to get vaccinated against COVID-19 on a yearly basis (online supplemental table 2). Similarly, there was a significant positive association between vaccine knowledge and psychological well-being with willingness to get yearly vaccines against COVID-19 (online supplemental table 2).

### RQ3 results
By far, the most important vaccine attribute for participants to choose a specific COVID-19 vaccine was effectiveness (57.2%, 95% CI 54.5% to 59.8%), while the least important was related to reputation of the vaccine based on the manufacturer and media coverage (9.7%, 95% CI 8.2% to 11.3%). All other vaccine attributes were rated

as similarly unimportant; vaccine technology: 13.3%, 95% CI 11.6% to 15.2%; side effects: 13.6%, 95% CI 11.9% to 15.5%; delivery method: 12.4%, 95% CI 10.7% to 14.2%; and ease of access: 13.4%, 95% CI 11.6% to 15.2%.

## DISCUSSION

In this sample of UK adults, vaccine hesitancy was negatively associated with receipt of/willingness to receive a first vaccine against COVID-19 and willingness to get a yearly COVID-19 vaccine. Vaccine knowledge was positively associated with receipt of/willingness to receive the initial vaccine against COVID-19 and willingness to get a yearly COVID-19 vaccine, while there was also a positive association between psychological well-being during the COVID-19 pandemic and willingness to get a yearly COVID-19 vaccine only. Effectiveness of the vaccine was the most important attribute for participants to choose a specific COVID-19 vaccine if they had the choice, while the least important was reputation of the vaccine based on the manufacturer and media coverage.

It should be noted that in the current study an overwhelming majority of respondents had either received or were willing to receive the first dose of COVID-19 vaccine, while vaccine hesitancy was low. Previous studies conducted in the UK have documented higher vaccine hesitancy and lower vaccine acceptance rates (between 63% and 69%).[31 32] However, most previous work was carried out in the early days of the COVID-19 vaccine roll-out, while our data were collected more than halfway through the vaccination programme in the UK. At that point of data collection, people might have had more confidence in the safety and effectiveness of COVID-19 vaccine, since not many side effects were reported from individuals who had already received the vaccine. It was also a period when COVID-19 measures were favourable for people who were vaccinated (ie, fully vaccinated did not have to quarantine when travelling back to England), which might have also influenced individuals' decision to get vaccinated in order to enable them to resume travelling and other activities and return to 'normal living' after a long period of social distancing measures and quarantines. Longitudinal and cross-sectional studies also have found that COVID-19 vaccination intention has increased over time in the UK,[7 33] with high compliance rates.[19]

The present study's high percentage of COVID-19 vaccine acceptance might also be attributed to the sociodemographic characteristics of the participants. For example, vaccine acceptance has been linked with being older, male, of white ethnicity and having higher qualification,[6–8] which reflects our sample characteristics. However, it should be noted that we weighted the sample to be representative of the population, so this cannot entirely explain the high acceptance rates observed.

Willingness to get vaccinated against COVID-19 on a yearly basis was also high, but lower than for the first dose of COVID-19 vaccine, and similar to previous research on US healthcare workers, which suggested an acceptance rate of a hypothetical annual booster vaccine against COVID-19 of 83.6%.[22] The lower levels of willingness to get a yearly COVID-19 vaccine compared with the first dose might be attributed to the fact that participants were less worried about a future wave of COVID-19, thus assuming they will not need protection against COVID-19 provided by the vaccine. Additionally, since people know that boosters are reserved for special populations (ie, all adults aged 50 years and over, healthcare professionals and those with a clinical risk[19]), they may think yearly vaccinations are unnecessary for themselves.

Vaccine knowledge was high in our sample, and it was positively associated with willingness to get vaccinated against COVID-19 initially and on a yearly basis. This may suggest that individuals who are better informed about vaccines in general have a greater understanding of the risk-benefit calculus of vaccinations, and thus of the benefits of the COVID-19 vaccine, specifically for individual and population health. Additionally, most of the participants had post-16 qualifications, and it has been documented that participants with a bachelor's degree are more likely to have a higher health literacy leading to a greater understanding of vaccine development and clinical trials to stop the spread of disease.[34] Earlier research also suggests that insufficient knowledge about the new COVID-19 vaccine and fears of long-term side effects were reasons not to get vaccinated. However, as more knowledge about the vaccine accumulated, and people become more educated about the vaccines, this may have increased willingness to accept the COVID-19 vaccine.[35–37]

Psychological well-being in our sample was average and positively associated only with willingness to get a yearly COVID-19 vaccine, and not receipt of/willingness to receive the initial COVID-19 vaccine. The average levels of psychological well-being reported by our sample may be due to the timing of data collection. High levels of psychological distress were documented during nationwide lockdowns and when infection levels and subsequent death rates were at their peak.[38] However, our study was conducted at a time when a new phase in the UK government's response to the pandemic was underway, moving away from stringent restrictions on day-to-day lives, towards advising people on how to protect themselves and others[39] and allowing for more socialisation. Additionally, an effective vaccine had already been developed and deployed. Previous research has found conflicting results about the association between psychological well-being and receipt of COVID-19 vaccine, with some findings linking poor psychological well-being, high levels of anxiety and worry with willingness to get vaccinated against COVID-19,[11 12] and others reporting significant association between poor psychological well-being and high levels of vaccine hesitancy.[13] Our results also suggest that people with higher levels of psychological well-being were more willing to get vaccinated against COVID-19 on a yearly basis. It could be argued that people with better psychological well-being may be less prone to worry/be anxious when it comes to trying new products/

vaccine, and thus more willing to get vaccinated against COVID-19.

Consistent with previous studies,[30 40] our findings indicate that vaccine efficacy is one of the most important determinants for COVID-19 vaccine acceptance. From a practical perspective, promotional materials could emphasise COVID-19 vaccine efficacy as a strategy to enhance the willingness for COVID-19 vaccine uptake. Aside from dispelling misinformation about vaccines, it may also be helpful to provide individuals with more information on why a particular type of vaccine can be effective.

### Strengths and limitations

The study benefited from a large sample size, use of validated measures of vaccine hesitancy and vaccine knowledge, as well as from examining willingness to get vaccinated against COVID-19 during a declared pandemic, and at a time during the roll-out of the first COVID-19 vaccine programme. This improves the validity of findings and minimises recall bias as the study was conducted when vaccination uptake was being actively promoted by government and health promotion authorities, and participants were faced with the actual question on whether or not to get vaccinated.

The study also has several limitations, including the use of a convenience sample who may have participated in the study due to a higher interest in the pandemic than in the general population. This interest may also be related to a higher receipt/willingness to get vaccinated against COVID-19. Additionally, the majority of participants were of white ethnicity, which meant we could not look at other ethnicities separately, when ethnic disparities in vaccine uptake/hesitancy have been reported elsewhere.[41] All data were self-reported and thus susceptible to social desirability bias. However, our rates of receipt/willingness to receive the first dose of COVID-19 vaccine were similar to the UK government rates of the first dose of COVID-19 vaccine.[19] Additionally, receipt of/willingness to receive an initial vaccine against COVID-19 was assessed only among those who have been offered a vaccine against COVID-19 at the time of the 12-month survey, thus excluding some younger participants.

### CONCLUSIONS

In conclusion, our findings suggest that willingness to get vaccinated against COVID-19 in a sample of UK adults during the time of the vaccine roll-out was high, vaccine hesitancy was low, general vaccine knowledge was high and psychological well-being was average. There was a negative association between vaccine hesitancy and acceptance/willingness to receive the initial dose of a COVID-19 vaccine and a yearly booster, a positive association between vaccine knowledge and willingness to receive both initial and yearly doses of a COVID-19 vaccine, while there was also a positive association between psychological well-being during the COVID-19 pandemic and willingness

to get a yearly COVID-19 vaccine. Effectiveness of the vaccine was the most important attribute for participants to choose a specific COVID-19 vaccine, if they had the choice. Our findings suggest that improving knowledge about vaccines and emphasising vaccine efficacy could minimise vaccine hesitancy and enhance the willingness for COVID-19 vaccine uptake. Thus, tailored education on vaccine hesitancy along with communication strategies that highlight vaccine efficacy is crucial to improve vaccination rates during the COVID-19 pandemic and future outbreaks.

**Acknowledgements** We are grateful to all the participants who have been supporting our research. We thank the Public Health England, and particularly the members of the Behavioural Insights at the Public Health England for providing feedback on the survey wording. We thank the Public Health England, Cancer Research UK, local authorities, mayors' offices, as well as charities and other organisations in the UK, including the Asthma UK and British Lung Foundation Partnership, for supporting our recruitment campaign.

**Contributors** LS, AH and DK conceived and designed the study. DK analysed the data and wrote the first draft. LS, ES and AH provided critical revisions. All authors read and approved the submitted manuscript. DK is responsible for the writing of this manuscript, accuracy of the data and accepts full responsibility for the work and/or the conduct of the study, had access to the data, controlled the decision to publish and is the guarantor for the overall content.

**Funding** This project is partially funded by an ongoing Cancer Research UK (CRUK) Programme Grant to UCL Tobacco and Alcohol Research Group (PRCRPG-Nov21\100002) and by SPECTRUM, a UK Prevention Research Partnership Consortium (grant number MR/S037519/1).

**Disclaimer** The funders had no role in the design and conduct of the study; collection, management, analysis and interpretation of data; preparation, review or approval of the manuscript; and decision to submit the manuscript for publication. For the purpose of Open Access, the author has applied a CC BY public copyright licence to any Author Accepted Manuscript version arising from this submission.

**Competing interests** LS has received honoraria for talks, an unrestricted research grant and travel expenses to attend meetings and workshops from Pfizer, and has acted as a paid reviewer for grant-awarding bodies and as a paid consultant for healthcare companies.

**Patient and public involvement** Patients and/or the public were involved in the design, or conduct, or reporting, or dissemination plans of this research. Refer to the Methods section for further details.

**Patient consent for publication** Consent obtained directly from patient(s).

**Ethics approval** The study has been approved by the UCL Research Ethics Committee at the UCL Division of Psychology and Language Sciences (PaLS) (CEHP/2020/579) as part of the larger programme 'The optimisation and implementation of interventions to change behaviours related to health and the environment'. The authors assert that all procedures contributing to this work comply with the ethical standards of the relevant national and institutional committees on human experimentation and with the Helsinki Declaration of 1975, as revised in 2008. All participants provided fully informed consent. The study is GDPR compliant.

**Provenance and peer review** Not commissioned; externally peer reviewed.

**Data availability statement** Data are available upon reasonable request. Data are available upon reasonable request from the lead author.

**ORCID iDs**
Dimitra Kale http://orcid.org/0000-0002-8845-7114
Aleksandra Herbec http://orcid.org/0000-0002-3339-7214

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
