## [Reviewer comments · BMJ Open]

ARTICLE DETAILS

TITLE (PROVISIONAL)	Willingness to get vaccinated initially and yearly against Covid-19 and its association with vaccine hesitancy, vaccine knowledge and psychological well-being: a cross-sectional study in UK adults
AUTHORS	Kale, Dimitra; Shoesmith, Emily; Herbec, Aleksandra; Shahab, Lion

VERSION 1 – REVIEW

REVIEWER	Basa, Muluken Trinity College Dublin
REVIEW RETURNED	30-Oct-2023

GENERAL COMMENTS	Abstract: The abstract is generally accurate but requires some modifications to align with BMJ guidelines. Please ensure that the abstract adheres to the specified structure and summarizes the study's key findings, methods, and implications more concisely. Introduction: The introduction effectively introduces the study subject, but it appears somewhat lengthy and contains uncited sentences. It is advisable to condense the introduction and cite relevant sources for the statements made. Study Design: The study design is generally appropriate for addressing the research question. However, it would be beneficial to explicitly state the study design in the methods section. Although the study is cross-sectional, the presence of various data collection times and follow-ups creates an impression of a cohort or pre-post design. Clarifying the timing of data collection in the methods would improve the clarity of the study design. Statistics: The use of statistics appears appropriate, but it is essential to include a statement regarding the assessment of the statistical validity and reliability in the methods section. Additionally, based on the results presented in Supplementary Table 1, a statistically significant difference is observed in the baseline characteristics of the included and excluded samples. This raises questions about the representativeness of the sample. It is crucial to discuss how this issue was addressed in the study. References: The references used are generally appropriate, but it is advisable to include more up-to-date sources, especially in a rapidly evolving field like COVID-19 research. Conclusions: The conclusions drawn in the study are justified by the results, but it would be beneficial to elaborate on the
--

	implications of these findings for current and future pandemics. Consider discussing the clinical implications and recommendations based on the study's results, as this will enhance the relevance and impact of the research.
--	---

REVIEWER	Socias, Sergio CONICET Tucuman
REVIEW RETURNED	27-Dec-2023

GENERAL COMMENTS	In the work "Willingness to get vaccinated initially and yearly against Covid-19 and its association with vaccine hesitancy, vaccine knowledge and psychological well-being: a cross-sectional study in UK adults", Dimitra et al. conducted a comprehensive investigation into the interplay among vaccine hesitancy, vaccine knowledge, psychological well-being, and their direct influence on the acceptance of Covid-19 vaccines. Their study delved into the determinants affecting individuals' readiness to receive initial Covid-19 vaccines and their ongoing willingness for yearly uptake. Additionally, the research critically evaluated the significance of diverse vaccine attributes in the selection process of a specific Covid-19 vaccine. Likewise, the present work emphasized the importance of understanding psychological well-being in promoting vaccine acceptance, thereby contributing to public health efforts during the pandemic. The present work led to relevant findings that highlight the potential to reduce vaccine hesitancy and enhance Covid-19 vaccination by improving vaccine knowledge and emphasizing their efficacy. The objectives of each analysis are very clear and the conclusions are mostly well founded. In my opinion, although its limitations, this study provides valuable insights into strategies that can be adopted to address vaccines hesitancy in order to increase their acceptance, and could be relevant to guide clinical decision-making and public health policies related to Covid 19 vaccination. Great Job! Congratulations!
--

VERSION 1 – AUTHOR RESPONSE

Reviewer: 1

Mr. Muluken Basa, Trinity College Dublin, Arba Minch University

Thank you for your time reviewing our manuscript and for your thoughtful and constructive comments.

Comments to the Author:

Abstract: The abstract is generally accurate but requires some modifications to align with BMJ guidelines. Please ensure that the abstract adheres to the specified structure and summarizes the study's key findings, methods, and implications more concisely.

Thank you for this comment. We have added a section 'outcome measures', where we present the outcome measures of our study. There is no methods section, but only 'design', 'setting', 'participant' and 'outcome measures' as per BMJ guidelines. We believe that the results and the conclusions

(there is no implication sections as per BMJ guidelines) of the abstract are brief and include only the necessary information.

Introduction: The introduction effectively introduces the study subject, but it appears somewhat lengthy and contains uncited sentences. It is advisable to condense the introduction and cite relevant sources for the statements made.

Thank you for this comment. We reviewed the introduction and made changes to condense it as per your suggestion. We also added references to some uncited statements.

Study Design: The study design is generally appropriate for addressing the research question. However, it would be beneficial to explicitly state the study design in the methods section. Although the study is cross-sectional, the presence of various data collection times and follow-ups creates an impression of a cohort or pre-post design. Clarifying the timing of data collection in the methods would improve the clarity of the study design.

We have rephrased the beginning of the methods/study design section to address your comment.

'This is a cross-sectional study using data...' p. 8, line 176. We also mention in the same section that 'Baseline data collection occurred between April-June 2020, and follow-up surveys were administered at 1 month, 3, 6 and 12 months from the baseline participation date. The current study uses data from the 12-month follow-up survey (supplemented by basic demographic information provided at baseline', p.8, lines 180-183. While in the study sample section, we provide exact dates of data collection for the 12-month follow-up '...12-month follow-up survey collected between May-June 2021. At the time of the 12-month follow-up survey, all clinically extremely vulnerable individuals and those aged +40 had been offered at least one dose of Covid-19 vaccine in UK', p. 9, lines 196-199.

Statistics: The use of statistics appears appropriate, but it is essential to include a statement regarding the assessment of the statistical validity and reliability in the methods section. Additionally, based on the results presented in Supplementary Table 1, a statistically significant difference is observed in the baseline characteristics of the included and excluded samples. This raises questions about the representativeness of the sample. It is crucial to discuss how this issue was addressed in the study.

Thank you for highlighting this, we have added Cronbach's alpha values of the instruments used in the methods section (p. 11, lines 253-254, p.12, lines 261-262). We also agree that our sample is not representative, and we have addressed this as a limitation of the study (p. 21). Additionally, as mentioned in the analysis section (p. 13, lines 287-290), data were weighted to the proportions of sex, age, ethnicity, education, and country of living obtained from the Office for National Statistics (2018) for the descriptive analysis to account for the non-random nature of the sampling.

References:

The references used are generally appropriate, but it is advisable to include more up-to-date sources, especially in a rapidly evolving field like COVID-19 research.

We appreciate this comment, and we updated some of the citations previously used (i.e., Kafadar et al., 2023; Kim et al., 2023; Yang et al., 2022; Galanis et al., 2022).

Conclusions: The conclusions drawn in the study are justified by the results, but it would be beneficial to elaborate on the implications of these findings for current and future pandemics. Consider discussing the clinical implications and recommendations based on the study's results, as this will enhance the relevance and impact of the research.

Thank you for this comment. We rephrased our conclusions and added the following 'Our findings suggest that improving knowledge about vaccines and emphasizing vaccine efficacy could minimise vaccine hesitancy and enhance the willingness for Covid-19 vaccine uptake. Thus, tailored education

on vaccine hesitancy along with communication strategies that highlight vaccine efficacy are crucial to improve vaccination rates during the Covid-19 pandemic and future outbreaks' p. 22, lines 499-503.

Reviewer: 2

Dr. Sergio Socias, CONICET Tucuman

Comments to the Author:

In the work "Willingness to get vaccinated initially and yearly against Covid-19 and its association with vaccine hesitancy, vaccine knowledge and psychological well-being: a cross-sectional study in UK adults", Dimitra et al. conducted a comprehensive investigation into the interplay among vaccine hesitancy, vaccine knowledge, psychological well-being, and their direct influence on the acceptance of Covid-19 vaccines. Their study delved into the determinants affecting individuals' readiness to receive initial Covid-19 vaccines and their ongoing willingness for yearly uptake. Additionally, the research critically evaluated the significance of diverse vaccine attributes in the selection process of a specific Covid-19 vaccine.

Likewise, the present work emphasized the importance of understanding psychological well-being in promoting vaccine acceptance, thereby contributing to public health efforts during the pandemic.

The present work led to relevant findings that highlight the potential to reduce vaccine hesitancy and enhance Covid-19 vaccination by improving vaccine knowledge and emphasizing their efficacy.

The objectives of each analysis are very clear and the conclusions are mostly well founded.

In my opinion, although its limitations, this study provides valuable insights into strategies that can be adopted to address vaccines hesitancy in order to increase their acceptance, and could be relevant to guide clinical decision-making and public health policies related to Covid 19 vaccination.

Great Job!

Congratulations!

Thank you for your time to review our manuscript and your kind words.